# Antagonising Chromatin Remodelling Activities in the Regulation of Mammalian Ribosomal Transcription

**DOI:** 10.3390/genes12070961

**Published:** 2021-06-24

**Authors:** Kanwal Tariq, Ann-Kristin Östlund Farrants

**Affiliations:** Department of Molecular Biosciences, The Wenner-Gren Institute, The Arrhenius Labs F4, Svante Arrhenius väg 20C, Stockholm University, SE-106 91 Stockholm, Sweden; kanwal.tariq@su.se

**Keywords:** rRNA gene repeats, chromatin states, NuRD, B-WICH, NoRC, histone modifications, non-coding RNA

## Abstract

Ribosomal transcription constitutes the major energy consuming process in cells and is regulated in response to proliferation, differentiation and metabolic conditions by several signalling pathways. These act on the transcription machinery but also on chromatin factors and ncRNA. The many ribosomal gene repeats are organised in a number of different chromatin states; active, poised, pseudosilent and repressed gene repeats. Some of these chromatin states are unique to the 47rRNA gene repeat and do not occur at other locations in the genome, such as the active state organised with the HMG protein UBF whereas other chromatin state are nucleosomal, harbouring both active and inactive histone marks. The number of repeats in a certain state varies on developmental stage and cell type; embryonic cells have more rRNA gene repeats organised in an open chromatin state, which is replaced by heterochromatin during differentiation, establishing different states depending on cell type. The 47S rRNA gene transcription is regulated in different ways depending on stimulus and chromatin state of individual gene repeats. This review will discuss the present knowledge about factors involved, such as chromatin remodelling factors NuRD, NoRC, CSB, B-WICH, histone modifying enzymes and histone chaperones, in altering gene expression and switching chromatin states in proliferation, differentiation, metabolic changes and stress responses.

## 1. Introduction

Ribosomal genes, both the nucleolar 47S genes and the 5S rRNA genes, are highly transcribed and contribute to the major part of all transcripts. The expression is tightly coupled to proliferation and cell growth with a high demand for protein synthesis [1,2,3] in mammalian cells. Therefore, ribosomal gene transcription is often dysregulated in highly proliferating cancer cells and is a target for novel cancer drugs [2,3,4,5,6]. Ribosomal transcription is regulated by environmental cues, such as changes in nutrient and energy supply, and thereby adjusts transcription and energy consumption [1,7,8]. The two ribosomal genes are transcribed by their own transcription machineries: 47S rRNA genes by RNA polymerase I (RNA pol I) and 5S rRNAs by RNA pol III. In addition, ribosomal transcription encompasses the transcription of ribosomal protein-coding genes by RNA pol II which is coordinated with the transcription by RNA pol I and RNA pol III [9]. Although the ribosomal genes are regarded as house-keeping genes recent studies have identified cell-type-specific expression, giving rise to tissue specific ribosomes with varying activity and specify [10,11]. Variations in ribosomal biogenesis, in particular in rRNA modifications, also contribute to the heterogeneity of ribosomes [12,13]. 

Ribosomal biogenesis takes place in the nucleolus, a membrane-less nuclear body that forms around active Nuclear Organising Regions (NORs), after each mitosis [14]. The NORs comprise gene clusters of the rRNA gene organised head to tail separated by an intergenic spacer region (IGS) [1,7,15]. Transcription of the genes, the processing and cleavage of the 47S rRNA transcript and the subsequent assembly of ribosomal RNAs and protein subunits occur in specific internal nucleolar structures dedicated to the different processes. The fibrillar centre (FC) assembles around active rRNA gene repeats associated with the RNA pol I factor upstream binding factor (UBF), the gene transcription occurs in the interface between the FC and the dense fibrillary centre (DFC), where processing and cleavage of the 47S rRNA take place, followed by the assembly of the rRNAs with ribosomal proteins to ribosomal subunits in the granular centre (GC) [16,17,18,19]. The formation of the nucleolar structures depends on transcriptional activity and when inhibited or very low, nucleolar structures are disrupted and nucleolar proteins are delocalised; fibrillarin, which is responsible for the 2′-O-methylation of rRNA early in the FC [6,20], and UBF translocate to the periphery, forming nucleolar caps (Figure 1A). Other abundant nucleolar proteins, such as the multifunctional proteins nucleolin, involved in activating rRNA gene transcription [21,22], and nucleophosmin (NPM), mainly involved in ribosomal assembly and export [22,23], are also leaving the nucleolus upon transcriptional inhibition and stress. The formation of the different nucleolar centres and the integrity of the nucleolus are based on phase separation and each centre contains many RNAs and several RNA binding proteins with intrinsically disordered domains [24]. Fibrillarin associates co-transcriptionally with the 5’-end of the growing rRNA chain in the FC and forms phase separated condensates representing DFCs [25]. Nucleolin and NPM are also involved in phase separation; nucleolin retains histone H2B in the nucleolus in a pre-rRNA dependent manner [26] and NPM is involved in protein quality control by forming phase separation condensates with misfolded proteins in the GC [27]. 

The nucleolus is also the target of several stress responses, such as heat shock, hypoxia, and DNA damage, all of which affect the expression of the 47S rRNA transcription [1,7,8]. In addition, nucleolar structure and function are disturbed in many viral infections to channel the resources to promote viral replication [27]. Many viruses accumulate in the nucleolus and bind to nucleolar proteins such as nucleolin, NPM and fibrillarin, to interfere with their function and 47S rRNA gene expression [28]. The many functions that congregate in the nucleolus and impact the 47S rRNA gene expression link nucleolar function closely to the status of the cell. The nucleolus constitutes a hub for responses, hence the 47S rRNA transcription is regulated by environmental states and stress on several levels; modification of the RNA pol I machinery and the chromatin landscape as well as altering nucleolar morphology and structure to alter the expression level. This review will focus on the regulation of 47S rRNA gene repeats in mammalian cells. 

### 1.1. The Active 47S rRNA Gene Transcription Is Organised in an Open Chromatin State

The human haploid genome harbours approximately 200 gene repeats found in clusters located on five acrocentric chromosomes, chromosome 13, 14, 15, 21 and 22 [7,8]. Not all genes are active in differentiated cells and some gene repeats are gradually silenced by the establishment of heterochromatin during differentiation [7,29,30,31]. The number of active gene repeats correlates to the requirement for protein synthesis in each cell but since the 47S rRNA transcription also needs to be adjusted according to cellular state and environmental conditions, several regulatory mechanisms exist also in differentiated cells. During differentiation, the gene repeats acquire different chromatin states with different compactions to meet changes in the cell. At least four different states, from open active gene repeats and to closed repressed gene repeats, have been identified (Figure 1B) [32,33,34,35,36,37]. 

The active 47S rRNA gene repeats are organised with the high motility group (HMG) protein UBF, which binds at the at the enhancer element, UCE, in the promoter, but also in the gene body and the upstream enhancer region [31,33,35,37]. UBF binds to UCE as a dimer covering approximately 140 bp of DNA in a single turn [35,38], and interacts with the TBP-containing TAFI-SL1 complex which binds at the core promoter at the transcription start site [39,40]. The UBF and SL1 recruit the RNA pol I bound to the axillary factor TIF-1A/RRN3 to form the pre-initiation complex (PIC) (Figure 2) [39,40,41]. UBF establishes an open chromatin state specific to the active 47S gene repeats [42] and is associated with GC-rich DNA at approximately every 170 bp in the gene body [37]. A further promoter [43], the spacer promoter binding UBF and SL1, is located approximately two kb upstream of the transcription start [35,37]. RNA pol I and TIF-1A/RRN3 are also found at the spacer promoter and a non-coding RNA (ncRNA), the spacer RNA, is produced in mouse [44,45,46]. Upstream of the spacer promoter is the boundary element, binding CTCF and cohesin, separating the nucleosomal IGS from the active UBF associated gene repeat [35,37,47,48,49]. The enhancer boundary element is in turn flanked by three to four positioned nucleosomes containing H3K4me2/3 and H2A-Z, indicating that the boundary constitutes an insulator (Figure 2). Similar enrichment of RNA pol I factors has been found in human cells approximately 800 bp upstream of transcription start site, with an enrichment of CTCF [37,48]. In the human IGS, a third promoter two kb upstream of the transcription start site in human cells was identified, from which an antisense RNA of unknown function is produced [50]. The open structure organised with UBF is particularly abundant in ES cells (embryonic stem cells) and mouse embryonic fibroblasts (MEFs), where few histone marks are found in the 47S rRNA gene repeat (35, 37), whereas that of differentiated cells may contain more histones (49). Interestingly, malignant human hepatocytes, HepG2, exhibit an increase of histone H3K4me3 approximately at the spacer promoter compared to control cells, and this is reduced when UBF is silenced.

### 1.2. The Boundary Element, a Form of Insulator, Is Important in the Maintenance of Chromatin States and Factor Loading

The boundary element functions as an insulator between chromatin configurations and may also represent a nucleation site for assemble factor that are responsible for switching chromatin structures (Figure 2). Several line of evidence support such an organisation: interfering with the composition of the element by siRNA silencing of CTCF impairs the loading of UBF on the 47S gene repeats [47] allowing for nucleosomes entering the gene repeats [32,34,35,36,42,49]. The boundary element and the adjacent active histones are present also in repressed gene repeats, suggesting that this site is a “marked” site and involved in establishing the active chromatin state [31,35,37]. Furthermore, the boundary element is enriched in chromatin remodelling ATPase SNF2h and the histone acetyl-transferase (HAT) p300, both of which are involved in altering chromatin configurations. The organisation of gene repeats with UBF is not sufficient for transcription, some of the active repeats are not engaged in transcription [35,37], and other factors, such as transcription factors, may be necessary for proper PIC formation to induce transcription.

### 1.3. Gene Expression of Active Gene Repeats Requires Transcription Factors

Initiation and elongation of the 47S rRNA transcription are rate limiting steps in responses to environmental cues without changing the chromatin state [42,51,52,53]. Enhancing elongation is the mauclein regulatory step in response to acute growth factor stimulation when the PIC is still in place, possibly by phosphorylation of UBF in the gene body [52,54]. Growth factor stimulation for cells in G1/G0, for instance after long-term, chronic starvation, requires PIC formation, RNA pol I loading as well as elongation regulation [52]. None of these regulatory steps require a change in chromatin states. This is further supported by the activation of 47S rRNA gene transcription of the proliferative transcription factor c-MYC in pre-malignant murine B-cells that occurs without a change in open UBF gene repeats organised with UBF [53]. Taken together, this suggest that the decondenced chromatin state organised with UBF forms the basis for transcription, as the competent state [53,55] (Figure 1B, panel 1), which is activated by specific transcription factors and by post-translational modification (PMT) of general factors. 

### 1.4. Active Gene Repeats Form Loops

Directly upstream of the UCE in the promoter, a binding site, T0, for transcription termination factor 1 (TTF-1) is located [56], and a similar site, Tsp, is also found at the spacer promoter, located 40 to 50 bp downstream of the UCE in mouse and human cells (Figure 2) [25,37]. Several TTF-1 sites, the Sal-boxes or T1-T10, are located at the end of the rRNA gene repeat to terminate the 47S rRNA transcription and it is also a replication fork barrier [57,58]. The function of the gene upstream sites is possibly to terminate ncRNAs produced upstream of the promoter, but the spatial organisation promotes interactions to form loops [59,60,61]. The spatial organisation of the active gene repeats in the nucleolus studied by super-resolution imaging techniques shows that RNA pol I forms ring structures with a size suggesting that active genes are organised in loops of one or a few genes being loaded with UBF or nucleosomes [25,62]. These loops may represent the loops between TTF-1 identified with 3C-techniques [59] or loops with CTCF [60,63]. It is not clear if the loops are formed between the promoter and the terminator of the same gene or proximal gene repeats or in what chromatin state these gene repeats are present, but the looping makes transcription more efficient.

### 1.5. Acetylation of UBF Enhances 47S rRNA Gene Transcription

Similar to histones, UBF is acetylated by histone acetyl-transferases (HATs), particular p300/CBP, and deacetylated by histone deacetylases (HDACs) to regulate the 47S rRNA expression [64,65]. Ribosomal transcription is altered through the cell cycle and UBF-Ac is higher in G2 when the transcription also is highest [66]. CBP acetylates UBF in S-phase to increase the interaction with PAF53, a RNA pol I associated factor, resulting in an increased 47S rRNA expression. Lack of acetylation of UBF also contributes to the reduced 47S rRNA transcription observed in a human Huntington disease model, suggested to be caused by the mutated Huntingtin protein sequestering the CBP protein [67]. Other HATs have been identified among processing factors and hALP and 16A/DRIM, t-UTP U3 processing factors, acetylate UBF which increases its interaction with RNA pol I [68,69]. 

## 2. The Inactive 47S rRNA Gene Repeats Are DNA Methylated and Organised in Constitutive Heterochromatin

The 47S rRNA gene repeats are maintained in and open chromatin state early in development, and then subsequent silencing of gene repeats is an important process during differentiation [70]. In the early embryo, most rRNA gene repeats are loaded with UBF and actively transcribed to meet the requirement of protein synthesis in the rapidly dividing cells [34,36,70,71] (Figure 1B, panel 1). The chromatin remodelling complex NoRC, composed of TIP5 (TIF interacting protein 5), the ATPase SNF2h, HDAC1, the histone methyl-transferase (HMT) SUV39H1, the DNA methylase DNMT3b is instrumental in establishing heterochromatin at the rDNA to reduce the 47S rRNA gene transcription [72]. It is also involved in establishing heterochromatin outside of the nucleolus, on centric and peri-centric repeats as well as major and minor repeats and at LINE elements [73]. The NoRC complex contributes to the restructuring of the nuclear architecture during embryogenesis by silencing active chromatin regions at repetitive regions. 

The repressed state established by NoRC is characterised by repressive chromatin marks, heterochromatin factors, and methylated DNA [72,74,75,76] (Figure 1B, panel 4). The complex is recruited to the promoter by TIP5 binding to TTF-1 [56,77], histone H4Ac [76,78,79] and linker DNA [80]. At the promoter, NoRC moves a promoter-bound nucleosome to a “closed” position incompatible with the assembly of the pre-initiation complex [76,80,81]. The closed promoter-bound nucleosome position (−132 to +22 bp) covers the core promoter and transcription start site, preventing SL1 from binding and initiating transcription. Subsequently the p-RNA, a fragment from the spacer-RNA, forms a triple helix with the T0 site, replaces TTF-1 and binds to TIP5 [45,46,75]. The p-RNA inhibits the ATPase activity of NoRC, which instead recruits the silencing factors HDACs, SUV39H1, and DNMT3 [81]. The conversion to heterochromatin involves the deacetylation of histones at the promoter, and the introduction of histone methylation, H3K9me2/3, and DNA methylation, which together reduces the binding of UBF [74]. The DNA-methylated promoter-bound nucleosome in quiescent cells also carries the histone modification histone H4K20me3 [82]. This histone modification is established by SUV420, and the protein is recruited to the rRNA promoter by the anti-sense transcript PAPAS (promoter and pre-rRNA antisense) [82]. This transcript is produced by RNA pol II in the antisense direction to the 47S rRNA, originates from the gene body but also includes reads through to the promoter, and is increased upon reduced levels of sense-transcription [82]. 

Histone variants of the canonical histones have been associated with rDNA. The DNA-methylated state binds macroH2A, which is associated with silent chromatin [83]. It accumulates on the rRNA gene promoter and maintains a silent chromatin state. Histone H1 is also associated with the silent gene repeats. Different H1 variants interact with silencing factors in the nucleolus, such as macroH2A, heterochromatin proteins 1α (HP1α), HP1β, and HP1γ and DNMT3a/b, but also with active factors, such as UBF, nucleolin and topoisomerase I [84,85,86]. Histone H1.2 interacts with UBF in mitotic NORs [87]. The binding of H1 variants also affect the rRNA gene transcription, where deletion of UBF [32] and Actinomycin D [88] increase H1 and HP1 association with the rRNA gene repeat. However, phosphorylated H1.4 co-localises with fibrillarin and actively transcribed rRNA [88]). Whether histone H1 and HP1 is functioning with NoRC is not clear but it does not exclusively associate with DNA methylated promoters, which suggests that it also is present in other chromatin states.

## 3. The Switch between Active and DNA-Methylated Inactive 47S rRNA Gene Repeats

The switch between active and DNA methylated inactive states involves the NoRC complex, which replaces UBF for nucleosomes in a process that occurs mainly during differentiation, embryonic development and malignant transformation. The switch employs specific transcription factors, chromatin remodelling and histone PTMs (Figure 1B, panel 1 and 4). The transcription factors recruit HDACs and SIRTs (NAD-dependent histone deacetylases) as well as histone HMTs, such as SUV39H1, either directly or as part of the NoRC complex to silence the genes [7,72]. The general transcription repressor BEND3 associates with the 47S rRNA and in its SUMOlated form interacts with NoRC and stabilises TIP5 by inhibiting its degradation through ubiquitination [89]. Furthermore, a number of bridging factors have been identified that respond to signalling pathways and affect the chromatin state. The activation of the 47S rRNA gene transcription by mTOR in human cell is counteracted by the tumour suppressor inhibitor of growth (ING1) [90]. ING1 is recruited to active rDNA by histone H3K4me3, and it recruits in turn the NoRC complex, which results in the establishment of heterochromatin and reduced 47S rRNA gene transcription. ING1 also inhibits the localisation of mTOR to the nucleolus. RUNX2, important in osteoblast differentiation, is not silencing rRNA gene repeats by NoRC, instead it binds the 47S rRNA gene promoters, recruit HDAC1 to deacetylase UBF, which leads to a decrease in RNA pol I transcription [91].

During development of certain lineages, transcription must be reactivated to allow for cellular transitions. This occurs in the epithelial-to-mesenchymal transition (EMT) in cells that need to develop a migratory cellular program, such as neural crest progenitor cells in early development. A temporary increased ribosomal transcription in G1/S-phase is required for EMT and is executed by the EMT-specific transcription factor Snail and the mTORC2 component Rictor [92]. The increase in transcription is associated with reduced DNA-methylation and replacement of TIP5 by UBF. A similar process is also found in cancer cells that rely on EMT for metastasis. It is not known if a chromatin-remodelling factor is involved in the antagonistic effect to NoRC in this process.

Little is known about the switch between rRNA repeats silenced with the histones variants macroH2A and H1 and active genes. If these gene repeats also exhibit further attributes of NoRC silenced chromatin than DNA-methylation is not known. Nevertheless, nucleolin counteracts the silent state imposed by macroH2A by increasing UBF loading along the rRNA gene body and increasing the presence of active histone marks at the promoter [83,93]. Nucleolin may act as a FACT-like H2A-H2B histone chaperon [94,95], removing macroH2A which leads to UBF loading, preventing TTF-1 binding to the T0 and the subsequent recruitment of TIP5 and NoRC [93]. In addition, inhibition of rRNA gene transcription results in higher levels of macroH2A at the 47S rRNA gene repeat, suggesting that both changes in nucleolin levels in the nucleolus and transcription leads to switches in chromatin state. Depletion of nucleolin affects the RNA pol I levels at the 5´end of the rRNA gene body, suggesting that not only transcriptional initiation but also elongation is affected. A similar effect is shown by the H2A-H2B histone chaperone FACT, which facilitate elongation through the gene body [96,97]. NPM/B23 also function as a histone chaperone at the promoter, requiring its RNA-binding domain and UBF to be recruited to the gene [98]. NPM inhibits histones and possible histone H1 [99] to associate with the 47S rRNA gene repeat and maintain and open chromatin structure [98]. Altering the binding of core histones and linker histones to the promoter or the gene body is one way to change the chromatin states of rRNA gene repeats and regulate gene expression. Nucleolar histone H1 interacts with both activating and repressing factors and is suggested to be involved in regulating different states. Several of the H1 isoforms have been localised to the nucleolus; H1.0, H1.2, H1.3 [86] and phosphorylated forms of H1.2 and H1.4 [88]. Phosphorylated histone H1.2 and in particular histone H1.4 associate with active 47S rRNA genes repeats and rRNA inhibition reduces the phosphorylation level. Histone H1 interacts with the histone chaperones nucleolin, NPM, subunits of FACT as well as the heterochromatic proteins macroH2A, and HP1 and the active proteins UBF, histone H2AZ, histone H3.3 and DNA topoisomerase II [84,86,87]. The interaction of histone H1 with several histone chaperones indicates that one function of the chaperones is to regulate histone H1 and UBF and their PMT at the promoter. Interestingly, H1 knock out gives more DNMT3a/b as well as proteins involved in chromatin remodelling, such as WSTF and BRG1, enriched in the nucleolus, indicating that chromatin changes and DNA methylation are involved in the regulation [85,100]. The changes caused by the histone variants and nucleolar chaperone proteins do not change the DNA-methylation fraction of gene repeats [49,83] which suggests that more chromatin states exist, constituted of nucleosomal chromatin on non-DNA-methylated 47S rRNA repeats.

### 3.1. The Permissive Poise Chromatin State Is Organised in Bivalent Chromatin on Unmethylated DNA

The rRNA gene promoters adopt further chromatin states than the active and repressed gene states referred to as the poised and pseudo-silent states, which in contrast to the repressed promoters silenced by NoRC, assemble on unmethylated DNA. The exact compositions of these states are largely unknown and they may constitute a group of differently organised nucleosomal gene repeats [31,37,49,101] with cell type specificity. The promoters carry UBF and SL1 as well as nucleosomes with bivalent chromatin marks [101] or histone H1 and HP1 [32,34]. The ChIP seq studies [33,35,37,49] of mouse and human rDNA suggest that the poised or pseudo-silent state is essentially nucleosomal and the active gene repeats with an open chromatin configuration are organised with UBF. In mouse, the poised state is associated with the two ATP-dependent chromatin remodelling complexes Cockayens syndrome protein B (CSB) and CHD4/ nucleosome remodelling deacetylase (NuRD), which act on the promoter nucleosomes (Figure 1B, panel 2 and 3) [101]. CSB and CHD4, the ATPase in the NuRD complex, are recruited to the promoter by TTF-1 during growth arrest and differentiation [101,102]. NuRD is responsible for the positioning of the promoter-bound nucleosome in the closed state covering the core promoter and transcription start site [101]. Although the position of the “closed” poised nucleosome is the same as that of the “closed” nucleosome formed by NoRC [81], the associating factors are different; the poised nucleosome carries both activate and silence histone marks, the active H3K4me3 and the two silencing H3K27me3 and H3K9me2/3, but no H4K20me3. It also differs from the NoRC “closed” nucleosome in that it is devoid of methylated DNA and associates with both UBF and SL1. Despite that UBF and SL1 are present at the promoter, TIF-1/RRN3 and RNA pol I are excluded [101], which shows that the poised closed nucleosome is not compatible with the assembly of the PIC. NuRD is suggested to prevent DNA methylation of the promoter, but still keep a poised pool of 47S rRNA gene repeats available for activation [101].In support of this, NuRD also inhibits TIP5 expression, indirectly contributing to inhibit the DNA methylation of rRNA repeats [103]. 

### 3.2. The Activation of the Permissive Poised State Involves Establishing an Active Nucleosome State

The poised “closed” marked nucleosome at the promoter is moved by the ATPase CSB to an active position (−157 to −2), compatible with TIF-1A/RRN3 and RNA pol I binding (Figure 1B, panel 2 and 3) [101]. CSB then recruits PCAF to the promoter to acetylate histone H4 and histone H3K9, creating an active configuration with histone H4Ac and histone H3K9Ac [104]. CSB associates also with the gene body, where it recruits the histone methyl transferase G9a to activate transcription by increasing the level of histone H3K9me2 and HP1γ in the region [102]. Recently, CSB together with CSA was shown to regulate transcription elongation mainly by recruiting nucleolin in human cells [105].

In addition to CSB, the poised state induced by the NuRD complex is counteracted by the ATP-dependent chromatin remodelling complex B-WICH (Figure 1B, panel 2 and 3). B-WICH, comprising WSTF, the ATPase SNF2h and nuclear myosin [106,107], activates rRNA gene transcription by establishing a more open chromatin configuration at the promoter to allow SL1, TIF-1A/RRN3 and RNA pol I to bind [108,109]. This state is characterised by the binding of UBF and a high histone H3K9Ac level at the promoter in human cells [108]. An impaired B-WICH complex, obtained by RNAi silencing of WSTF, results in CHD4 not being released from the promoter after glucose stimulation, suggesting that B-WICH is required to replace NuRD to open up the chromatin to allow the pre-initiation complex to form [109]. B-WICH also allows other factors, such as c-MYC, to bind to the promoter and enhance expression of the genes [109]. 

### 3.3. Transcriptions Factors, c-MYC, Induce Chromatin Changes and Activate Gene Expression

c-MYC is a transcription factor that regulate ribosomal and metabolic genes in response to proliferative stimulus and in complex with MAX binds to several position in rDNA, including a site in the 47S gene promoter [110,111]. NPM facilitates the nucleolar entry of c-MYC and its transcription activation [112]. c-MYC recruits the TRRAP co-regulatory complex that contain the HATs GCN5, TIP60 and PCAF, which acetylate histones in the promoters [113]. c-MYC is also involved in the induction 47S rRNA gene transcription in hypertrophic mouse muscle, where it associates with the promoter together with WSTF [114]. B-WICH opens up chromatin for c-MYC-MAX binding in the intergenic spacer between the 5S rRNA genes, which is a pre-requisite for histone modifications and transcription [115]. Along with regulating the ribosomal gene transcription directly, c-MYC regulates the expression of RNA pol I factors [116]. The regulation of c-MYC influences ribosomal transcription and some pathways are targeting both c-MYC and ribosomal factors; c-MYC and UBF are degraded following SUMOlation by PIAS E3 ligases [117] and c-MYC mRNA stability is coordinated with 47S rRNA gene transcription through the RNA guanine-7 methyltransferase in the mRNA cap [118].

c-MYC is dysregulated in cancer [119] usually with a concomitant increase in 47S rRNA gene transcription. This increase may be linked to a change in chromatin state at 47S rRNA gene repeats but may also result in other changes. In contrast to c-Myc expression in murine pre-malignant B-cells which have an enhanced rRNA transcription without chromatin changes, the malignant transformed cells have an increased number of active gene repeats, but no change in gene expression occurred [53]. Instead, a higher extent of interactions between UBF organised rRNA gene repeats and RNA pol II genes in the peri-nucleolar region appeared [53]. These UBF organised gene repeats were shown to display an enhanced interaction with RNA pol II gene enhancers in the peri-nucleolar region. The UBF has been shown to associate with a subset of highly expressed RNA pol II genes [120]. The change in chromatin state in malignant transformed cells, referred to as the rDNA class switching, does not involve changes in DNA methylation, most likely switching an unmethylated repressed pool of 47S rRNA gene repeats [53], such as the poised or pseudo-silent state. 

In addition to c-MYC, other factors, such as ING4, are involved in activating transcription by altering the chromatin configuration. ING4 activates transcription by inducing histone H3K9Ac and histone H4Ac and increasing the level of UBF at promoters [121]. It is not known whether these transcription factors activate 47S rRNA gene transcription by altering both active gene repeats and permissive poised repeats at the same time and by the same mechanism, recruiting chromatin remodellers or histone modifying complexes. 

### 3.4. HATs and HDACs Are Involved in Chromatin Changes

HATs, such as MOF, p300/CBP, PCAF, GCN5, TIP60, are associated with the switch to an active state introducing histone H3Ac and histone H4Ac [49,72,78,101,108], functioning with transcription factors, such a c-MYC. In tumour cells, overexpression of LYAR, a transcription factor in inflammatory pathways, also MYST/KAT7 bound to BRD2 and BRD4 associates to UBF to acetylate histone H4 and H3 [122]. Not only acetylation of histones and UBF is associated with activation; subunits in SL1 acetylated by PCAF [123], which are deacetylated by the NAD-dependent SIRT1 to inhibit transcription during mitosis [124]. SIRT7 is a further NAD-dependent deacetylase associated with ribosomal transcription, but unlike SIRT1, it is an activator [125]. SIRT7 localises to the nucleolus, it interacts with UBF and is required for the release from mitosis [126,127]. SIRT7 has many targets in the nucleolus affecting both transcription and processing that have been acetylated by CBS; PAF53, an RNA pol I associated factor, to activate transcription [128], the helicase DDX21 to resolve R-loop [129] and U3-155, NOP56 and fibrillarin, to promote processing and cleavage of the 47S rRNA [130,131]. SIRT7 also deacetylate fibrillarin at the exit of mitosis promoting methylation of histone H2A at glutamine 104 (H2AQ104), which decompacts the promoter to resume transcription [129,132,133]. Other active histone marks in RNA pol II transcription, such as H3K4me3 and H3K36me3, are also present 47S rRNA gene repeats, most likely at the permissive poised state [101]. These modifications are altered by the histone demethylases KDM2A/B, in response to glucose starvation and metabolites through AMPK pathway and HP1γ recruitments [134,135,136,137,138]. Two histone demethylases, PHD8 and KDM4B, are involved in switching to an active chromatin state by removing methyl groups from histone H3K9me3/2n [139], PHD8 in response to PIP2-binding [140]. The fact that many histone modifying enzymes associate with the promoter suggests that they modify nucleosomes in the permissive poised or pseudo-silent state and facilitate the switch to an active nucleosomal state.

### 3.5. eNOSC Induces Silencing in Response to Low NADH-Levels

A low energy level is sensed as an increased AMP/ATP ratio by the AMPK pathway, which downregulates several processes to save energy, among those the 47S rRNA gene transcription [141]. The AMPK inhibits mTOR which reduces the rRNA gene transcription by inhibitory phosphorylation of TIF-1A/RRN3 and UBF [142,143,144,145] to prevent the preinitiation complex from forming [146]. In HeLa cells, which lack the AMPK-pathway, a chromatin-remodelling complex, eNoSC, which senses a reduced energy level in the form of an increased NAD+ level has been identified [147]. The eNoSC, comprising nucleomethylin (NML), SIRT1 and SUV39H1, is recruited to the promoter by nucleosomes harbouring H3K9me2 and in turn establishes heterochromatin by deacetylation and methylation of histone H3K9 [147]. NML also binds to the 47S rRNA and associates with the promoter first when released after reduction in transcription [148,149]. The NML protein is also involved in rRNA methylation and function through p53 in senescence [150,151]. Since eNoSC only associates with the promoter after transcription is reduced it is most likely that the complex functions in the maintenance of a silent chromatin in the energy saving pathway rather than in establishing a silent chromatin state. In addition to eNOSC, NuRD establishes a repressed chromatin state in glucose starved cells, and we propose that B-WICH counteracts this chromatin structure upon glucose stimulation [109]. 

## 4. Acute Stress Responses and the Regulation of rRNA Transcription

The nucleolus acts as a hub for different stress responses, such as viral infections, hypoxia and heat shock, and these responses reduce the 47S rRNA gene transcription. Many stress responses are sensed by signals and targets the RNA pol I factors directly, which inhibit the assembly of the PIC but others are altering the chromatin states at the genes. Hypotonic stress [152] and heat shock [153,154] require the NuRD complex to establish a silent state at the promoter. SUV420, which is recruited by PAPAS during growth arrest to establish a compaction of the chromatin, is degraded by the E3-ligase NEDD4 during stress [152]. This leaves the PAPAS, which forms a triple helix with the enhancer element at the promoter, to bind to and recruit the dephosphorylated CHD4 in the NuRD complex to the promoter [154]. The mouse cells, the PAPAS is induced by heat shock and hypotonic stress by the simultaneous dephosphorylation of TIF-1A/RRN3 as CHD4 [153,155], and this results in reduced rRNA gene transcription and an increased PAPAS production.

### 4.1. lncRNA Originating from the Human IGS Are INVOLVED in Stress Responses with Reduced 47S rRNA Gene Transcription 

pRNA and PAPAS are not found in human cells, but a number of ncRNAs that originate from the IGS have been identified. These are associated with different stress responses; heat shock induces the expression of transcripts, IGS16, and IGS22, originating from loci 16 kb and 22 kb from the transcription start site and acidosis induces a transcript, IGS28, from a locus 28 kb, all of which in the sense direction [156,157]. These ncRNAs bind stress proteins, such as the chaperon HSP70 and the acidosis responsive von Hippel– Lindau (VHL) protein and sequester them from their site of action. DNMT1, POLD1, and RNA biogenesis factors, such as the RNA pol I and III factors RPA16 and RPA40, are other proteins that bind to these ncRNAs, further emphasising the role of these transcripts in sequestering and immobilising factors important in different pathways during stress [156,157]. Another long ncRNA, the PNCTR (pyrimidine-rich noncoding transcript), originating from 28 kb, is proposed to sequester the RNA binding protein PTBP1 to the peri-nucleolar centre, to interfere with splicing and prevent apoptosis [158]. This stress related RNA is overexpressed in cancers and suggested to form a scaffold for the peri-nucleolar centre, a structure which is larger in cancer cells. The shorter ncRNAs from the IGS also form scaffolds in phase separated condensates, liquid-like detention centres, with sequestered mobile proteins. These condensates disrupt the nucleolar structure and reduces 47S rRNA gene transcription [159,160,161]. In particular, IGS16, IGS22, and IGS23 contain low-complexity RNA structures formed by repetitive cytosine/uracil (CU) or adenosine/guanine (AG) sequences, which are proposed to immobilise proteins with cationic domain and fibrillation-propensity domains [160]. The transcription of these short ncRNA initiate the formation of phase-separated aggregates, which resembles solid-like amyloid bodies, in response to thermal stress and extracellular acidosis [160]. It has been suggested that these aggregates are under the surveillance of chaperones in the protein quality control to protect against nucleolar disruption [160,161]. 

SincRNA transcripts produced by RNA pol I have been identified from the same region in the human IGS as the stress-induced ncRNA [162]. Similar to the stress-induced ncRNAs, the sincRNAs disrupt nucleolar structure and reduce 47S gene transcription. The expression of sincRNAs is reduced by R-loops formed by RNA pol II antisense transcription at the same locus. The R-loops produced prevent RNA pol I from loading and transcribe and is regulated by the helicase Sentaxin, which facilitates RNA pol II loading at the IGS. R-loops are found at many location, which suggests that the whole region is transcribed and possibly is under control to reduce detrimental ncRNA transcription that can feed phase separation condensates [163,164]. 

### 4.2. Nuclear Integrity Is Important for 47S rRNA Gene Transcription 

Nucleolar integrity is tightly coupled to 47S rRNA gene transcription, with many studies even showing that the assembly of nucleoli at NORs depends on active transcription [6,16,17,18,19,20,165,166,167], and a reduced transcription results in in reorganised or dispersed nucleoli [18,19,24,165,166,168,169,170]. DNA damage and inhibited 47S gene transcription lead to a reorganisation of the nucleolar structure, with the typical nucleolar caps [166,168,171,172]. A region immediately outside of the rDNA locus on all five acrocentric chromosomes harbours conserved sequences, the Distal Junction (DJ), of about 400 kb. This region is located at the peri-nucleolar heterochromatin and anchors the rDNA [172]. It is actively transcribed by RNA pol II from at least four promoters, but the function of these transcripts is unknown. The DJ regions, perhaps their transcripts, enhance 47S rRNA gene transcription by promoting active NORs from different chromosomes to coalescence possible by promoting phase separation [167,173]. 

Intron-derived Alu-repeat sequences in human cells, or B1 elements in mouse cells, originating from RNA pol II genes in the nucleoplasm are also important for proper nucleolar assembly, possibly by promoting coalescence of individual small nucleoli driven by phase separation. Alu-repeat sequences bind to nucleolin, NPM and fibrillarin and are required for nucleolin and NPM pre-nucleolar bodies to form larger nucleolar structures after exit from mitosis [174]. Inhibition of RNA pol II results in dispersed nucleoli and the depletion of Alu-sequences reduces 47S rRNA gene transcription as a consequence of low levels of transcription factors in these smaller nucleoli [168,174].

The stress-related ncRNAs are specific for human cells, suggesting that the trigger for regulation of nucleolar integrity of nucleoli differs between species but the outcome is the same. In addition to these RNAs, further species-specific RNAs that regulate 47S rRNA gene transcription have been identified. The SLERT sno-containing RNA produced by alternative splicing from the TBRG4 locus affects RNA pol I transcription in human cells [175]. It is present in ES cells and cancer cells and increases 47S rRNA gene transcription by inhibiting the helicase DDX21. This RNA is not found in mouse cells, but regulation of the 47S rRNA gene transcription by ncRNAs occurs. Nucleolar lncRNAs have been identified from mouse brain, and the LoNa (nucleolar-specific lncRNA) binds nucleolin and fibrillarin and reduces 47S rRNA gene transcription [176].

## 5. The Trigger to Switch between Different States

Regulation of 47S rRNA gene transcription employ a network of factors and pathways, with several layers of cross-talk between the processes. Cellular states, such as proliferation, differentiation, metabolic changes and cellular stress conditions, regulate rRNA gene transcription by phosphorylation of the RNA pol I factors to change their interaction in the PIC formation [1,2,3,7,8]). Proliferation activates kinases in the RAS-MAPK pathway and PI3/AKT-kinase pathway as well as cyclin-CDKs (cyclin dependent kinases) during the G1 and S-phases, which phosphorylate UBF, SL1 and TIF-1A/RRN3 [177,178,179,180]. mTOR is induced in response to nutrient availability [142,143] and AMPK in response to energy starvation [146], and both pathways activate kinases that phosphorylate RNA pol I factors. In addition, cellular stress induces JNK2 [155] and PTEN [181,182] to inhibit the RNA pol machinery. These signalling pathways are most probable functioning on genes in an active chromatin state, and only stimuli requiring long-term changes in expression cause alterations of chromatin states. How long term external or internal signals are conveyed and the mechanism behind switching between chromatin states is less clear.

### TTF-1 Appears to Be a Determining Factor in the Switch

Many factors have been associated with the switch between the permissive poised state and the active state. In mouse cells, the NoRC, NuRD and CSB complexes bind to TTF-1 and are recruited to the promoter region where they remodel the chromatin [80,101]. TTF-1 is in itself an important factor in the activation of 47S rRNA gene transcription by forming loops between of active genes [59,60,61]. The level of TTF-1 in the nucleolus is changed upon DNA damage and oncogenic stress, when the tumour suppressor p19/14 ARF is induced and inhibits the nucleolar localisation of TTF-1 [183]. In addition, nucleolin reduces the TTF-1 bound to the promoter [93]. The lower level of TTF-1 at the rRNA gene results in a reduced 47S rRNA gene transcription [93,183], possibly by reducing the ability to form TTF-1 loops and induce heterochromatin formation. TTF-1-mediated increase of 47S rRNA gene transcription occurs in insulin-stimulated adipocytes by activation of the caveolar protein Cav-1/PTRF [184,185]. Cav-1/PTRF is phosphorylated upon stimulation, localises to the nucleus, binds the TTF-1 at the 47 rRNA gene repeat, and enhances the loop formation between the promoter and the termination of the rRNA gene [184]. In addition to TTF-1, other proteins in the rRNA gene repeat also are involved in loop formation, such as c-MYC, which is responsible for loops formed in response to transcription [186] and possibly CTCF although two inverted binding sites have not been identified [35,37,47,48,49]. These looping factors may compensate for each other, and siRNA silencing of TTF-1 in HeLa cells does not reduce c-MYC recruitment [109].

How TTF-1 regulate its interactions with chromatin remodellers and affects gene expression is not fully understood. Nucleolin reduces the binding of TTF-1 to T0, which decreases the association of TIP5 with the promoter [93] but whether this also affects CHD4 is not known. The interaction between TTF-1 and TIP5 in NoRC is further regulated by PTM; TIP5 is acetylated by MOF and this inhibits the interaction with TTF-1 and pRNA [187] and is deacetylated by SIRT1 upon glucose deprivation allowing for its interaction with TTF-1 to establish heterochromatin [75,81,187,188,189]. TTF-1 binding to CHD4 in mouse cells requires the nucleosomes at the promoter to be modified by the MLL/SET to contain histone H3K4me3 [101]. NuRD then establishes a bivalent mark consisting of histone H3K4me3 and histone H3K27me3/H3K9me3 on nucleosomes by promoting histone H3K4me3 formation, which affects the binding of other regulating proteins. The higher level of histone H3K4me3 counteracts the binding of SHPRH, a human protein that enhances 47S rRNA gene transcription by recruiting RNA pol I to the promoter [190,191]. The SHPRH is released from the promoter upon starvation or mTOR inhibition and binds instead to gene body, which correlates with a NuRD-dependent histone H3K4me3 increase at the promoter, and a simultaneous increase of histone H3K4me2 in the gene body. However, siRNA silencing of TTF-1 in HeLa cells suggests that TTF-1 is not necessary for targeting NuRD to the promoter; instead, NuRD enhances TTF-1 binding at the promoter [109]. B-WICH subunits, which counteract NuRD silencing, do not interact with TTF-1 directly, but the complex restrains the level associated with the promoter. A reduced level of TTF-1 results in an enhanced level of UBF at the promoter, suggesting that TTF-1 is involved in establishing or stabilising chromatin states at the promoter which in turn is regulated by the NuRD and B-WICH. This raises the question how the chromatin remodelling complexes are recruited to the promoter. B-WICH association with the promoter is independent of transcription, suggesting that the subunits interact with the chromatin at the promoter directly. In addition, WSTF is hyper-phosphorylated and stays at the rDNA throughout mitosis [192]. The WSTF also interacts with nucleolar proteins at the promoter, such CSB, DDX21 and the Myb-binding protein [107] and SIRT7 [193,194].

## 6. Conclusions

The 47S rRNA gene repeats are organised in several in different chromatin states and differently regulated to meet the need from the environment. In mammalian cells, the active genes are organised with UBF in an open chromatin state, constitutively repressed genes are heterochromatinised on methylated-DNA, and further gene repeats are in the less defined permissive poised state or pseudo silent state (Figure 1B). It is tempting to speculate that the UBF-organised gene repeats are regulated by PTMs of the different RNA pol transcription factors and by transcription factors, such as c-MYC; either to enhance or inhibit the formation of the PIC or change elongation rate in response to environmental conditions. During differentiation, active UBF-organised gene repeats are silenced by NoRC, which establishes a constitutively repressed nucleosomal state with the silent histone modifications H3K9me3, H4K20me3 at a DNA methylated promoter. In cancer, this repressed state is derepressed into an active state loaded with UBF, but how this switch is achieved is not known. It appears to require dysregulated oncogenes, such as c-MYC or Snail, binding at the promoter [52,53,92], but if this switch also occurs in non-malign cells is not clear. In addition to the active and the constitutively repressed state, a permissive poised state and a pseudo-silent state have been identified, possibly providing the cell with a pool of genes that can be switched on and off to preserve homeostasis in response to metabolic states. These chromatin states appear to be heterogeneous but represents a nucleosomal state at low-methylated DNA promoters that carries bivalent histone modifications and chromatin proteins. Different chromatin remodellers and factors, such as NuRD or histone H1, are associated with the permissive silent poised state and are counteracted by activating complexes, CSB [101] and B-WICH [108,109]. How these factors are recruited and regulated are poorly understood, but TTF-1 and ncRNAs appear to be involved, in addition to possible PMTs of the chromatin remodelling factors. Histone chaperones, nucleolin, NPM, and FACT, also function to regulate the chromatin state on the 47S rRNA gene repeat, but it is unclear if they work to clear histones from UBF-associated active repeats, which has been suggested for nucleolin [83], or if they are involved in a switch between chromatin states on nucleosomal repeats. Which chromatin states present in each cell may be cell type dependent and rely on which factors that are expressed, WSTF in B-WICH is highly expressed in neural crest cells [195].

Several ncRNA from both the rDNA and outside of the nucleolus are involved in the regulation of 47S rRNA gene transcription. These work in two ways, either by influencing the chromatin structure at the promoter or by binding proteins influencing the structure of the nucleolus (Figure 1A,B). These mechanisms are general and found in most species. Inhibited transcription results in a reorganisation of the nucleolar structure and the fibrillarin and UBF form nucleolar caps both in mouse and human cells [167,196]. However, the ncRNAs that changes the chromatin structure, pRNA or the PAPAS, have still to be identified in human cells. In human cells, ncRNAs are formed in response to different stress conditions to form phase separated condensates and disperse the nucleoli [156,166,170], which leads to a reduce transcription. The differences between mouse and man may reflect differences in the architecture of the IGSs and the induction of ncRNAs but the ability and mechanisms to respond to environmental cues and stress is most likely the same. Nevertheless, we have identified ncRNAs that affect the accessibility of the promoter and the spacer promoter and regulate the 47S rRNA gene transcription originating from the region 30kb to 40 kb from the transcription start site of the human IGS (Tariq and Östlund Farrants, unpublished results), indicating that also human ribosomal gene expression relies directly on chromatin changes imposed by RNA. Still many questions remain about the composition of different chromatin states and the mechanisms behind chromatin switches. Investigations has been hampered because of technical difficulties in studying repetitive gene and to distinguish between chromatin states of individual 47S rRNA gene repeats. With new imaging techniques and transcriptomics and proteomics techniques we will obtain new insights into the nature of ribosomal gene regulation.

## Figures and Tables

**Figure 1 genes-12-00961-f001:**
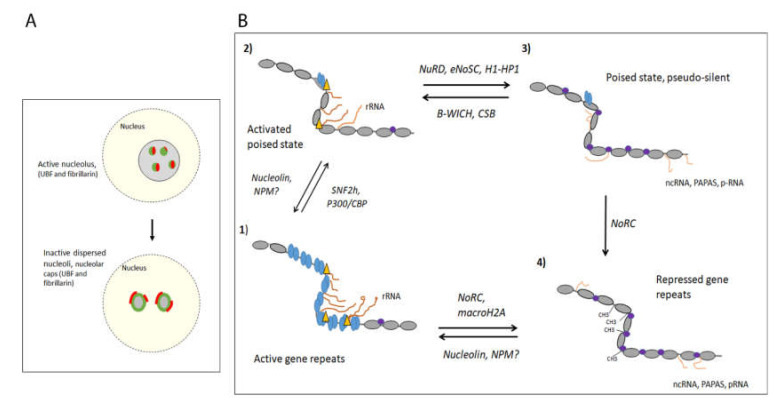
(**A**) The nucleolar integrity in active and dispersed nucleoli. Nucleolus with several FCs reflecting active transcription, with UBF (green) and fibrillarin (red) assemblies surrounded by GC. (top panel), and nucleus exposed to stress such as DNA damage or starvation, reducing transcription, leading dispersed nuclei with a different organisation, here shown with nucleolar caps, UBF (green) and fibrillarin (red) at the periphery and GC-component in the interior (bottom panel). (**B**) Different chromatin states of the mammalian rDNA and the chromatin remodelling complex identified. (1) The active chromatin state is organised with UBF (bleu ovals), which imposes an open, decondensed chromatin state that is transcriptionally active if induced by transcription factors (RNA pol I is marked with a yellow triangle), with nucleosomal chromatin (grey ovals) in the IGS; (2) the activated poised state, organised with nucleosomes (grey ovals) and is transcriptionally active (RNA pol I is depicted with a yellow triangle, (3) the permissive poised state or the pseudo-silent state, organised with nucleosomes carrying histone PMTs and heterochromatin factors (marked with purple circles), and transcriptionally inactive, (4) the constitutively repressed chromatin state, organised in heterochromatin with DNA-methylations and is transcriptionally silent. The chromatin remodelling complexes and histone chaperones involved in the switches between states are depicted around the arrows. RNAs are shown, the sense 47S rRNA transcript in (1) and (2) and ncRNA species from the IGS in (3) and (4).

**Figure 2 genes-12-00961-f002:**
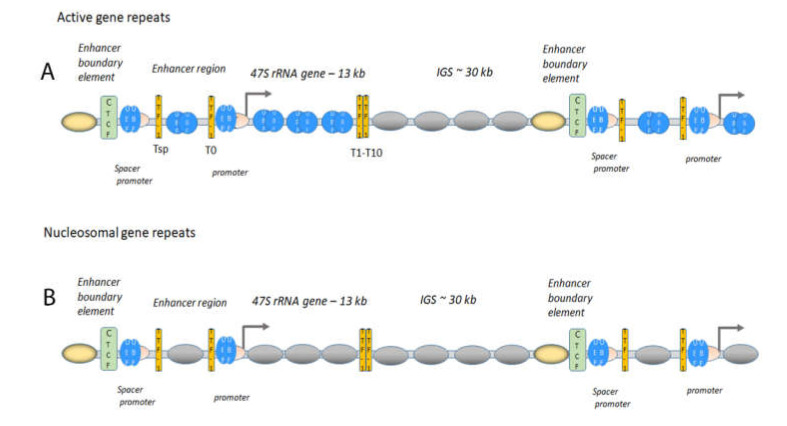
Chromatin architecture at the rDNA at the active (**A**) and the poised state (**B**). (**A**) The genes are organised head-to-tail; UBF (bleu ovals) and SL1 (pink crescent) are bound to the promoter and the spacer promoter (depicted in the figure). TTF-1 (yellow bars) is bound to its binding sites, depicted Tsp downstream of the spacer promoter, T0 at the promoter, and T1-T10, at the end of the 47S rRNA gene. UBF is associated with the active 47S rRNA gene body and with the upstream enhancer region and the IGS downstream of the termination of the gene (T1-T10) is nucleosomal (grey ovals). CTCF (green rectangle) is flanking the spacer promoter with positioned nucleosomes with active histone marks and H2AZ (yellow oval nucleosome). These actively marked nucleosome are also present in the poised state in the bottom panel). (**B**) The poised state have nucleosomes (grey ovals) in the gene repeat, UBF (bleu ovals and SL1 (pink crescent) at the promoter and spacer promoter. TTF-1 (yellow bars) is binding to its sites, Tsp, T0, T1-T10 in the gene repeat. CTCF (green rectangle) is also here flanking the spacer promoter with positioned nucleosomes with active histone marks and H2AZ (yellow oval nucleosome).

## Data Availability

Not applicable.

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
