# Peer review of "Antagonising Chromatin Remodelling Activities in the Regulation of Mammalian Ribosomal Transcription"

_genes, 2021, doi:10.3390/genes12070961_

Round 1
Reviewer 1 Report
The authors present a review on rDNA chromatin structure and the dynamic switching of chromatin states during up- or downregulation of rRNA synthesis. The topic has received revived interest lately in the context of cancer biology and in developmental and aging studies. Thus, the presented paper is timely of interest for the community.
The review covers the field in a rather complete way and offers an up-to date citation of the contemporary literature. Apart from quite some typos I have one general remark and several smaller and more specific remarks.
General remark: What is generally lacking is the view on histones, in particular histone replacement, which seems important for a review on switching chromatin states. The authors could consider to add a sub-chapter on that.
Specific remarks:
Line 40: when explaining what NOR means I suggest to use „Region“ instead of „Center“ in order to make the “R” clear.
Line 81: You rightly refer to UBF here but I would suggest to also introduce fibrillarin and nucleolin. These two factors are dealt with later in the review so it might be helpful to mention them with a few lines here.
Figure 1: It would be helpful for the reader to have explanation of some of the elements. For me, e.g. it is unclear what the yellow triangle should indicate. Please make this clear in the legend, maybe also indicating the other shapes/colors.
Figure 2: The scheme is very helpful in general but I have troubles to understand the rose-colored ellipses to which UBF binds. The elliptical shape seems to imply that this is a nucleosome maybe different from the other, greyish nucleosomes. The legend does not explain what it is; if this should indicate the promoters I suggest to use another shape to avoid confusion or clearly explain in the legend. SL1 is mentioned in the legend but not in the scheme; please add that it is not shown. Please also reformulate the sentence in line 110, I guess it should explain that these nucleosomes carry active marks.
Line 339: 5S?
Line 392: I guess it should read “NucleOLar”?
Line 507: There is a problem with format of the literature.
Figure 3: I have problems with the central part of the scheme including the dashed lines pointing towards it. First of all, it is unclear what exactly red and green is. Supposing red is UBF and green is fibrillarin: which data/studies lead the authors to conclude that fibrillarin has a ring-like distribution around nucleoli in dispersed nucleoli. The scheme also implies that only in this case poised genes exist (dashed arrow). Also, one would interpret the scheme such that there would not exist poised and inactive repeats in active nucleoli, which is certainly not the case. Additionally, to my knowledge it has not been shown that the activated, poised genes are located at or within the fibrillar complexes as is implied in the scheme. I strongly suggest to revise Figure 3 to avoid confusion.
Author Response
Dear editor,
We have now revised the manuscript genes-1252836 “Antagonising chromatin remodelling activities in the regulation of mammalian ribosomal transcription” (and added mammalian in the title) according to the reviews’ comments. The reviews made several excellent suggestions how to improve the manuscript and we have tried to accommodate them all.
Below follows a point-to-point reply:
Review:
General remark: What is generally lacking is the view on histones, in particular histone replacement, which seems important for a review on switching chromatin states. The authors could consider to add a sub-chapter on that.
We have now included histone chaperones and replacement in many sections were it fitted, and also changed the figures accordingly.
Specific remarks:
Line 40: when explaining what NOR means I suggest to use „Region“ instead of „Center“ in order to make the “R” clear. This is now changed
Line 81: You rightly refer to UBF here but I would suggest to also introduce fibrillarin and nucleolin. These two factors are dealt with later in the review so it might be helpful to mention them with a few lines here. We have now included fibrillarin, nucleolin and nucleophosmin.
Figure 1: It would be helpful for the reader to have explanation of some of the elements. For me, e.g. it is unclear what the yellow triangle should indicate. Please make this clear in the legend, maybe also indicating the other shapes/colors. We now explain the figure elements and the figure legends are rewritten.
Figure 2: The scheme is very helpful in general but I have troubles to understand the rose-colored ellipses to which UBF binds. The elliptical shape seems to imply that this is a nucleosome maybe different from the other, greyish nucleosomes. The legend does not explain what it is; if this should indicate the promoters I suggest to use another shape to avoid confusion or clearly explain in the legend. SL1 is mentioned in the legend but not in the scheme; please add that it is not shown. Please also reformulate the sentence in line 110, I guess it should explain that these nucleosomes carry active marks. We now explain the figure elements and the figure legends are rewritten.
These changes are in line with the reviewer's comments which has improved the manuscript.
Kind regards
Anki Östlund Farrants, corresponding author

Reviewer 2 Report
The review genes-1252836 proposed by Tariq and Östlund Farrants, provides interesting information concerning molecular mechanisms controlling rRNA chromatin regulation and expression in eukaryotic cells. The review is mostly addressed to findings in human cells but it is likewise a good source of information and inspiration for laboratories working in similar topics in other organisms.
The review is clearly written and refer to well-known but also recently published data. My major concern are the figures that can be improved and eventually reorganized.
1) Figure 1 too small,
2) in same figure is NuRD not NURD
3) Indicate in the legend: UBF dimer are blue circles, 45S rRNA are in orange, violet circler are ? and yellow triangles …..etc
4) In Figure 2 check CTCF and TTF1 font sizes and not clearly annotated
5) In Figure 2: indicate T0 and Tsp sites mentioned in lanes 142-143
6) Figure 3 is quite similar to Figure 1 (redundant?). It Should be re-made but focused in the nucleolar structures
7) Lane 43: “the transcription and processing of the 47S rRNA transcript occurs into the DFC”. To my understanding these early evens occurs at the interphase of FC and DFC. While later processing occurs in the DFC. This point need to be more precisely explained. At least if this is the case in mammals.
8) Lane s160-163: “Acetylation of UBf also……..the CBP protein”, this sentence is not clear
8) Lane 339: …after reduction of 4S? you mean 47S
9) Page 505: “These mechanism are general and found in most species”. Need to be mention which species and provide reference?
10) Lane 513: “…we have identified non-codong RNA “. You mean non-coding?
11) It would be nice if authors give more detail concerning these non-coding RNA from IGS
12) Tthe authors should address the question if chromatin factors as NuRD, NoRC…are evolutionary conserved among different species.
Author Response
Dear editor,
We have now revised the manuscript genes-1252836 “Antagonising chromatin remodelling activities in the regulation of mammalian ribosomal transcription” (and added mammalian in the title) according to the reviews’ comments. The reviews made several excellent suggestions how to improve the manuscript and we have tried to accommodate them all.
Below follows a point-to-point reply:
Reviewer 2
Line 339: 5S? corrected
Line 392: I guess it should read “NucleOLar”? corrected
Line 507: There is a problem with format of the literature. Corrected
Figure 3: I have problems with the central part of the scheme including the dashed lines pointing towards it. First of all, it is unclear what exactly red and green is. Supposing red is UBF and green is fibrillarin: which data/studies lead the authors to conclude that fibrillarin has a ring-like distribution around nucleoli in dispersed nucleoli. The scheme also implies that only in this case poised genes exist (dashed arrow). Also, one would interpret the scheme such that there would not exist poised and inactive repeats in active nucleoli, which is certainly not the case. Additionally, to my knowledge it has not been shown that the activated, poised genes are located at or within the fibrillar complexes as is implied in the scheme. I strongly suggest to revise Figure 3 to avoid confusion. This figure is removed and made into Fig1A and Figure 1B.
1) Figure 1 too small, Figure 1 is reorganised and made larger
2) in same figure is NuRD not NURD corrected
3) Indicate in the legend: UBF dimer are blue circles, 45S rRNA are in orange, violet circler are ? and yellow triangles …..etc . Rewritten
4) In Figure 2 check CTCF and TTF1 font sizes and not clearly annotated. Changed Figure 2 according to the reviewer’s comment appropriate e descriptions.
5) In Figure 2: indicate T0 and Tsp sites mentioned in lanes 142-143
6) Figure 3 is quite similar to Figure 1 (redundant?). It Should be re-made but focused in the nucleolar structures. We have removed Figure 3.
7) Lane 43: “the transcription and processing of the 47S rRNA transcript occurs into the DFC”. To my understanding these early evens occurs at the interphase of FC and DFC. While later processing occurs in the DFC. This point need to be more precisely explained. At least if this is the case in mammals. This point has now been clarified and rewritten.
8) Lane s160-163: “Acetylation of UBf also……..the CBP protein”, this sentence is not clear. This is now rewritten.
8) Lane 339: …after reduction of 4S? you mean 47S. Corrected
9) Page 505: “These mechanism are general and found in most species”. Need to be mention which species and provide reference? We have now changed this and only mention mouse and man.
10) Lane 513: “…we have identified non-codong RNA “. You mean non-coding? Corrected
11) It would be nice if authors give more detail concerning these non-coding RNA from IGS. We have now specified the location
12) Tthe authors should address the question if chromatin factors as NuRD, NoRC…are evolutionary conserved among different species. This is not included, since we have focused on the mammalian cells, and this we have cleared now by adding mammalian in the title and a sentence that we focus on mammalian 47S in the introduction. To address the evolutionary aspect would take focus away from the main chromatin question.
We hope that we have addressed the queries satisfactory,
Kind regards
Ann-Kristin Östlund Farrants
Corresponding autho
